# Image-Based Annotation of Chemogenomic Libraries for Phenotypic Screening

**DOI:** 10.3390/molecules27041439

**Published:** 2022-02-21

**Authors:** Amelie Tjaden, Apirat Chaikuad, Eric Kowarz, Rolf Marschalek, Stefan Knapp, Martin Schröder, Susanne Müller

**Affiliations:** 1Institute of Pharmaceutical Chemistry, Goethe University Frankfurt, Max-von-Laue-Str.9, 60438 Frankfurt, Germany; tjaden@pharmchem.uni-frankfurt.de (A.T.); chaikuad@pharmchem.uni-frankfurt.de (A.C.); knapp@pharmchem.uni-frankfurt.de (S.K.); 2Structural Genomics Consortium, BMLS, Goethe University Frankfurt, Max-von-Laue-Str. 15, 60438 Frankfurt, Germany; 3Institute of Pharmaceutical Biology, Goethe University, Max-von-Laue-Str.9, 60438 Frankfurt, Germany; kowarz@em.uni-frankfurt.de (E.K.); rolf.marschalek@em.uni-frankfurt.de (R.M.)

**Keywords:** phenotypic screening, high content imaging, chemogenomics, machine learning, cell cycle

## Abstract

Phenotypical screening is a widely used approach in drug discovery for the identification of small molecules with cellular activities. However, functional annotation of identified hits often poses a challenge. The development of small molecules with narrow or exclusive target selectivity such as chemical probes and chemogenomic (CG) libraries, greatly diminishes this challenge, but non-specific effects caused by compound toxicity or interference with basic cellular functions still pose a problem to associate phenotypic readouts with molecular targets. Hence, each compound should ideally be comprehensively characterized regarding its effects on general cell functions. Here, we report an optimized live-cell multiplexed assay that classifies cells based on nuclear morphology, presenting an excellent indicator for cellular responses such as early apoptosis and necrosis. This basic readout in combination with the detection of other general cell damaging activities of small molecules such as changes in cytoskeletal morphology, cell cycle and mitochondrial health provides a comprehensive time-dependent characterization of the effect of small molecules on cellular health in a single experiment. The developed high-content assay offers multi-dimensional comprehensive characterization that can be used to delineate generic effects regarding cell functions and cell viability, allowing an assessment of compound suitability for subsequent detailed phenotypic and mechanistic studies.

## 1. Introduction

Phenotypic screening has recently experienced a resurgence in drug discovery after many years of focus on target based approaches [1]. In particular, methods such as cell painting [2,3,4] or phenomics are gaining interest due to their ability to detect disease relevant morphological and expression signatures. These exciting new technologies provide insights into the biological effects of small molecules on cellular systems and the suitability of identified hits for translational studies. One of the main advantages of phenotypic screening lies in the potential of identifying functionally active chemical modulators without the need to know their precise mode of action (MoA). However, the lack of detailed mechanistic insight complicates the rational development of identified hit matter and validation studies [5]. One way to circumvent these complications is the use of better annotated chemical libraries, consisting of highly target-specific chemical probes [6,7,8] or chemogenomics libraries which contain well-characterized inhibitors with narrow but not exclusive target selectivity [9,10]. In particular, the latter have gained increasing interest as a new approach in drug discovery [11,12] as chemogenomic libraries may cover a large diversity of targets and a larger fraction of druggable proteins. Thus, chemogenomic compounds (CGCs) can supplement chemical probe collections that are not available for many targets due to their stringent quality criteria [13]. In cellular studies, the use of several CGCs directed towards one target but with diverse additional activities, will allow deconvolution of phenotypic readouts and identification of the target causing the cellular effect [14,15]. In addition, compounds from diverse chemical scaffolds may enable an easier identification of off-targets from different families. Further validation such as proteomic-based approaches or quantitative structure-activity relationships (QSAR) may be required [16]. The importance of chemogenomics for drug development has recently been demonstrated by a call of the Innovative Medicines Initiative (IMI), which resulted in the funding of the EUbOPEN project. One aim of this project is to assemble an open access chemogenomic library covering more than 1000 proteins by well annotated CGCs as well as chemical probes [17]. The expansion of this CGC collection to cover the entire druggable proteome will be the goal of the Target 2035 [18].

While target selectivity is an important parameter, there is a need for a comprehensive annotation of CGCs in terms of quality of the used chemical matter, such as structural identity, purity and solubility. In addition, the effects of CGCs on basic cellular functions such as cell viability, mitochondrial health, membrane integrity, cell cycle and interference with cytoskeletal functions which may be affected by non-specific binding of CGCs to tubulin should be considered [19]. Computational as well as screening approaches have been employed to predict the (unspecific) toxicity of libraries used for screening [20]. Although it is not always easy to distinguish between on-target and off-target effects in a cell viability assay, adding information on chemical and biological quality to CGC libraries will help to differentiate between target specific and unspecific effects [10]. New technology developments such as automated image analysis systems and machine learning algorithms enabled high-content techniques to become the method of choice for the essential annotation of chemogenomic libraries. Here, we present a modular live-cell high-content cellular viability assay, which we expanded to include assessment of CGC effects on the cell cycle, tubulin, mitochondrial health and membrane integrity. In contrast to the Cell Painting assay, which captures a multitude of morphological features of fixed cells at a given time point and requires extensive downstream analysis [3], the purpose of our assay is to describe cell health in living cells, providing the opportunity for real-time measurement over a long time-period. The modular nature of the assay offers the opportunity for an expansion such as adding compound-safety assays and other cellular stress reporter systems without the need for additional informatics capacities [21].

## 2. Results

### 2.1. Optimization of HighVia Protocol and Validation of Cell Staining Dyes

Analyzing cytotoxicity at multiple time points improves the annotation of small molecules and facilitates distinguishing between primary and secondary target effects. In this report, we improved our previously published, single time point protocol [22] to provide a more continuous readout. Live-cell imaging using fluorescent dyes for an extended period of time requires low concentrations of dyes that do not interfere with cellular functions yet provide fluorescent signals that are sufficiently high for robust detection. Therefore, we first optimized the concentration of the DNA-staining dye Hoechst33342 and determined 50 nM as the minimal concentration that still yielded robust detection of nuclei in HeLa cells (Appendix A). Previous studies have identified the toxicity level of nuclear stains such as Hoechst33342 at concentrations around 1 µM [23,24]. We tested in a cell viability experiment using U2OS cells and the alamarBlue™ dye (alamarBlue HS reagent, ThermoFisher, Massachusetts, United States) whether Hoechst33342 at concentrations below 170 nM resulted in reduced viability (Figure 1A/Appendix A). Additionally, we assessed in this experiment the potential effects on cell viability of other live-cell dyes such as the mitochondrial stain MitotrackerRed^®^ and the taxol-derived tubulin dye BioTracker™ 488 Green Microtubule Cytoskeleton Dye. Gratifyingly, none of the dyes exerted any significant impairment of cell viability at the proposed assay concentration over a time period of 72 h (Figure 1A). To exclude the potential influence of multiple dyes at their given concentration would influence viability, we tested different combinations of dyes in U2OS cells using an orthogonal high content readout (Figure 1B/Appendix A). Consistent with the data of the single dye experiments, none of the dyes or their combination inhibited cell viability.

Encouraged by these results, we assessed whether by lowering the dye concentrations of the previously published HighVia protocol (workflow of HighVia protocol see Figure 1C), this method could be adapted to allow a continuous readout (workflow continuous protocol see Figure 1D). In addition to the dyes used in the HighVia protocol, we included MitotrackerDeepRed^®^ to measure the mitochondrial content and thus named the protocol ‘HighVia Extend’. Changes of mitochondrial mass are indicative of certain cytotoxic events such as apoptosis [25,26]. Cells were detected as previously described [22] and gated into five different populations using a supervised machine-learning algorithm (Figure 1E/Appendix A. We chose nine reference compounds as a training set (Appendix A) for the assay setup, which was tested in three different human cell lines: human embryonic kidney cells (HEK293T), osteosarcoma cells (U2OS) and non-transformed human fibroblasts (MRC9). These reference compounds covered multiple modes of actions including topoisomerase inhibitor camptothecin, which triggers apoptotic cell death by inducing strand breaks in chromosomal DNA [27], the BET bromodomain inhibitor JQ1 [28], the mTOR inhibitor torin [29] and the glycosidic drug digitonin, a detergent used to permeabilize cell membranes [30] (Figure 1F). We found that the new continuous assay format captured the kinetics of the selected diverse cell death mechanisms: the cell-membrane permeabilizing agent digitonin as well as the multikinase inhibitor staurosporine and the ATM/ATR inhibitor berzosertib displayed rapid induction of cytotoxicity, while inhibitors of epigenetic targets JQ1 and ricolinostat showed slower and less pronounced cytotoxic effects (Figure 1F), consistent with previous reports [31,32]. Treatment with the non-selective CDK inhibitor milciclib, the mTOR inhibitor torin and the tubulin-disassembly inhibitor paclitaxel resulted in cytotoxic response with intermediate kinetics. IC_50_ values of the different compounds over time are compiled in Appendix A.

Consistent with the overall cell count of healthy cells, the population gating also followed different kinetic profiles, exemplified by camptothecin (Figure 1G). These data suggested that the continuous format of the HighVia Extend facilitated the assessment of time-dependent cytotoxic effects of small molecule compounds.

### 2.2. Investigation of Nuclear Properties

While analyzing data from the continuous experiment, we identified a strong correlation between the overall cellular phenotype (categories: “healthy”, “early/late apoptotic”, “necrotic”, “lysed”) and the nuclear phenotype defined as either “healthy”, “pyknosed” or “fragmented” (Figure 2A). To further test whether the gating based on the nuclear phenotype alone resulted in similar cytotoxicity profiles, we compared the calculated IC_50_ values of the aforementioned nine reference compounds gated either as described above (based on entire cellular phenotype) or based on the nuclear phenotype alone (features used for machine learning algorithm described in Appendix A). We found that the time-dependent IC_50_ values and the maximal reduction in the healthy cell population were highly comparable between these gating methods (Figure 2B,C/Appendix A). Additionally, the overall population distribution profiles from both gating protocols were highly similar (Figure 2D,E). The dependence on only one fluorescent channel might however increase the risk of assay interference of compounds with similar fluorescent properties such as berzosertib or with poorly soluble small molecules such as itraconazole that exhibit high fluorescent background (Figure 2F, Appendix A). In order to minimize the risk of such interferences, we included an additional layer of gating to the protocol. In the first step, all fluorescent objects in the corresponding channel were classified either as “nuclei”, independent of their phenotype or as “high intensity objects” that detected both fluorescent compounds as well as precipitations (Figure 2G,H). We noticed that a limitation of this method was that we were not able to readily distinguish between pyknosed nuclei of mitotic cells or condensed nuclei of apoptotic cells. However, performing a normalization of the healthy nuclear count against the healthy nuclear count of the DMSO controls (see Materials and Method section) eliminated the uncertainty between mitotic and apoptotic nuclei and corrected the overall information on the healthy nuclear count per well.

These data suggested that the classification of cells based on their nuclear phenotype can be used as a surrogate of more complex gating protocols. Thus, the gating based on the Hoechst33342 signal simplified the High-Content assay setups by not only enabling the counting and identification of cells but also by assessing their health state and compound properties such as intrinsic compound fluorescence or the occurrence of compound precipitation.

### 2.3. FUCCI Cell Cycle Analysis

Validating effects of small molecules on the cell cycle is an important test for new drug candidates which is frequently assessed, e.g., by Fluorescence-activated cell sorting (FACS) using DNA-binding dyes such as propidium iodine (PI) [33] or the cell cycle dependent degradation of fluorescent maker proteins described by Sakaue-Sawano et al. [34]. The FUCCI technology allows us to distinguish between live cells in different cell phases using a dual-color imaging system [35]. Thanks to the opposing effects of the licensing factor Cdt1 (RFP-tagged) and its inhibitor Geminin (GFP-tagged) on DNA replication, their presence, seen by the fluorescent tag, can be used to distinguish between G1 and S/G2/M phases of the cell cycle. Cells in S/G2/M are identified by a GFP-labeled nucleus (hereafter referred to as “green”). Cells in G1 result in RFP-labeled nuclei (“red”) and those in the transition state between G1 and S phase, show both GFP and RFP-labeled nuclei (‘yellow”). A small fraction of non-labeled nuclei that appear shortly in between M and G1 phase is rare and can be neglected in the analysis [36].

To test the compatibility of the nuclei-based gating protocol in combination with other fluorescent markers, we used this cell cycle reporter together with the described nuclear gating strategy in HCT116 cells. We chose this cell line for this experiment due to its favorable nucleus:cell-body distribution.. In our analysis, we focused on the cell cycle phases of unaltered, ‘healthy’ gated nuclei, but also the pyknosed, and to a certain extent, the fragmented populations could be further gated based on the intensities of the FUCCI reporters (general workflow see Figure 3A). We only considered the three major populations of “green”, “red” and “yellow” nuclei, while the neglectable fraction of non-labeled cells was excluded (Figure 3B).

We first assessed whether the introduction of the FUCCI system would influence sensitivity of cells with respect to compound viability. HCT116-FUCCI cells treated with the CDK inhibitor milciclib resulted in a comparable cytotoxic profile as observed with the same protocol above for U2OS cells (Figure 3C). The gating based on the nuclear phenotype enabled the exclusive analysis of cells not showing an apoptotic or damaged phenotype over several time points in one experiment. Comparing the effect of HCT116 treated with 1 µM of milciclib with the one of DMSO treated cells resulted in the expected alteration of cell cycle distribution upon compound treatment. Milciclib treated cells not only showed a lower number of “healthy” classified cells but also displayed a higher population of ‘red’ nuclei after 18 h of compound treatment (Figure 3D), indicative of a G1 phase arrest, consistent with milciclib’s ability of inhibiting cyclin-dependent kinases such as CDK2 [37,38]. The timing of the accumulation of cells in the G1 phase after 18 h correlated well with the less pronounced cytotoxicity at the earliest time point of 3.5 h, further pointing to a primary rather than secondary compound effect. These data underlined the advantages of simultaneously detecting cytotoxicity and investigating different phenotypes at several time points in one experiment.

Encouraged by these results, we analyzed the effect of additional 17 compounds at a single concentration of 10 µM in HCT116-FUCCI cells for up to 70 h (Figure 3E). This set of compounds included compounds known to affect the cell cycle, such as the topoisomerase II inhibitors daunorubicin, doxorubicin as well as mitoxantrone and topoisomerase I inhibitors camptothecin and topotecan. We also included paclitaxel and vinorelbine tartrate, representing drugs that influence the tubulin function. Overall, the compounds represented a broad range of mechanisms affecting the cell cycle, such as triggering check point response, and various cytotoxic agents. A full list of known effects of the used compounds is included in Appendix A.

The population analysis confirmed an increase in red nuclei for the flavone derivate α-naphthoflavone [39], the tubulin binding taxol-derivative paclitaxel [40,41] and MEK1/2 inhibitor trametinib [42] consistent with previous reports of these compounds to cause a cell cycle arrest in G1. An increase in green nuclei, in comparison to DMSO, was detected for both tested topoisomerases I inhibitors, camptothecin and topotecan, as well as mitoxantrone, which are known to cause mitotic cell cycle arrest [27,43,44,45,46]. More yellow nuclei were observed for cells treated with the topoisomerase II inhibitors daunorubicin and doxorubicin, both of which are known to cause cell cycle arrest due to DNA double strand breaks [44,47]. Treatment of cells with the TOPK inhibitor HI-TOPK-032 [48] also resulted in an increased number of cells with yellow nuclei, indicating induction of an S phase arrest. In general, all compounds that have been described to interfere in cell cycle progression, showed the expected effects in the FUCCI assay system. Figure 3F shows an example of the nuclei population analysis for the three compounds, trametinib, mitoxantrone and HI-TOPK-032 in comparison to cells exposed to DMSO 0.1%. Additional data can be found in Appendix A.

### 2.4. Multiplex Protocol

Low cell viability can be the result of on-target effects, off-target effects or be based on undesirable characteristics of compounds that interfere with the assay system. We therefore included the assessment of tubulin structure, mitochondrial mass changes and membrane permeability in the phenotypical analysis. We used a test set of 21 compounds with known effects on cell viability to validate the protocol (Appendix A). However, we envisioned that depending on the phenotypes of interest, additional cell staining dyes can be used to detect further changes in cellular morphology or health. Compounds that modulate microtubule functions have been used extensively in cancer research, because of their interference with tumor growth. However, unspecific or unrecognized tubulin binding can lead to false interpretation of presumed target specific effects, in particular in oncological assays. We therefore included a microtubule cytoskeleton dye to detect changes in the tubulin structure. In addition, mitochondrial health was monitored using Mitotracker™ Red (Invitrogen), providing information on mitochondrial mass, which has been shown to provide a good indicator for the apoptotic susceptibility of cells. Higher mitochondrial mass can lead to cell damage, because mitochondria are the main source of ROS (reactive oxygen species) [26]. Membrane permeability was detected using the microtubule cytoskeleton dye and cells were compared to digitonin, as a cell lysis control. Cellular shape and fluorescence were measured as before, at 12 h and 24 h after compound treatment. To analyze effects on tubulin, mitochondria and membrane permeability, a machine learning-based protocol was implemented, based on four earlier tested compounds as a training set for the algorithm (Appendix A). Cells that showed different tubulin appearance compared to the DMSO 0.1% control were marked as ‘tubulin effect’. Cells that showed an increase in mitochondrial mass in comparison to DMSO 0.1% were marked ‘mitochondrial mass increase’ and cells that showed membrane permeability were marked ‘membrane permeable’. For membrane permeability, the reference compound digitonin, a cell detergent leading to cell perforation, was used. In our first approach, we gated all cells, independently of their viability, into the different phenotypical groups, namely ‘tubulin effect’ or ‘no tubulin effect’. With this first analysis, (Appendix A) we saw that an increase in mitochondrial mass as well as a tubulin effect was frequently associated with cytotoxicity. When adding ‘cytotoxic’ compounds such as staurosporine (10 µM) or puromycin (10 µM), most cells showed as expected increased mitochondrial mass and tubulin effects [25]. To exclude these phenotypic effects caused merely by cell death, only cells defined as ‘healthy nuclei’ were gated into the different groups. The generic workflow of the analysis is shown in Figure 4A. Compounds that have certain quality deficiencies, shown by not passing the property thresholds, were marked. First, compounds that showed a Hoechst High Intensity Object ratio of more than 50% (Figure 4B) were selected, because they either precipitated or showed auto-fluorescent, interfering with the assay readout.

For the test set of 21 compounds (Appendix A), two compounds (camptothecin and topotecan) showed Hoechst High Intensity Objects at levels > 30% after 24 h in one of two biological replicates at 10 µM. Image validation demonstrated, that depending on the location of the precipitated compound, the intensity levels of the channels could vary, thus both duplicates should be considered as precipitation is a stochastic event and dependents on compound handling [49]. All other “normal” cells were then gated based on their nuclei properties in “healthy”, “pyknosed” or “fragmented” (Figure 4C). As mentioned above, a distinction between cells that showed condensed nuclei while undergoing apoptosis and cells that were mitotic was only possible taking into account the total cell number and comparison to DMSO as a control. To increase the robustness of this parameter, we added Annexin V as an apoptotic marker to the Hoechst33342 stain. After inclusion of this marker, it was possible to distinguish between mitosis and apoptosis for U2OS cells. Even for the human embryonic kidney cells (HEK293T), which are smaller and rounder than the other cell lines used, the distinction between mitosis and apoptosis was confirmed by normalization to DMSO as control. For the tested small chemogenomic set, all compounds that showed less than 50% of healthy cells were marked and should be assessed further. In our test set, five compounds were marked at 10 µM (daunorubicin, staurosporine, topotecan, camptothecin and puromycin), as expected. To detect phenotypical properties, that are independent of cell death, only nuclei that were gated healthy earlier, were considered. Vinorelbine tartrate, a vinca alkaloid with antimicrotubule properties interfering with mitotic spindle function, was used as a positive control for tubulin effects [50]. As expected, after 24 h, vinorelbine tartrate treated cells, exhibited increased tubulin effects compared to DMSO 0.1% treated cells. For mitochondrial mass increase, we used milciclib as a test compound. Milciclib, is a known apoptosis modulator [51] and showed an increase in mitochondrial mass over 60% in comparison to DMSO. The chemical probe compounds, SR-318 (a chemical probe for MAPK14) [52] and BAY-179 (a chemical probe for complex I) also showed over 80% healthy nuclei but an increase in mitochondrial mass over 60%. The MAPK14 probe SR-318 as well as the orthogonal dual MAPK14/DDR probe SR-302 also showed a tubulin effect of more than 90%. Interestingly, compounds that are known to permeabilize the membrane such as digitonin (10 µM), showed still more than 50% healthy nuclei after 24 h. The same compounds were tested in HEK293T and MRC-9 cells. The data have been included in the Appendix A.

The tested compounds were used, to establish a protocol for detection of multiple readouts. The following requirements were determined for compound flagging: Hoechst High Intensity Objects > 50%, Healthy Nuclei < 50%, Pyknosed Nuclei > 50%, Fragmented Nuclei > 50%, Tubulin effect > 50%, mitochondrial mass increase > 50%, membrane permeability > 50%. However, we recommend that every experiment should contain control compounds with known characteristics as an internal standard. This primary screen was able to flag compounds, which should be further investigated, regarding their suitability for inclusion as compounds in a chemogenomic set. For instance, target specific and off-targeted mechanisms affecting cellular health can be distinguished by control compounds and/or alternative inhibitors with diverse chemical structure that makes it unlikely that these targets also inter with similar off-target mechanisms as the investigated compound.

### 2.5. Multiplex Analysis of Chemogenomic Compounds

To validate whether this assay can be performed in a medium throughput format, we tested a small library of 215 compounds at two different concentrations, 1 µM and 10 µM, in U2OS, HEK293T and MRC-9 cells. Here, we describe the analysis of the U2OS cells while the results for the further two cell lines are provided in the Appendix A. Most of the compounds tested have cellular on-target activities in the nanomolar range. It was thus not surprising to observe strong viability effects for a large number of compounds at 10 µM (Figure 5A). We therefore mainly evaluated the effects at the lower concentration of 1 µM. In the first step, compounds that showed Hoechst High Intensity Objects were considered as described earlier. Here, only three of the 215 compounds showed more than 50% of Hoechst intensity compared to normal nuclei after 1 µM treatment: the control compounds berzosertib and camptothecin, as well as the FGFR inhibitor PD173074 [53], which precipitated, a property that can be explained by its hydrophobicity (logP of 4.7). It should be noted that PD173074 has a cellular activity at less than 25 nM and should therefore be used at lower concentrations than 1 µM. The validation of cell viability using the nuclei gating described earlier, revealed 20 compounds with less than 50% of healthy nuclei. Among these, only nine compounds (volasertib, BMS-754807, DDR-TRK-1N, TP0903, GNF-5837, infigratinib, adavosertib, ML154, omipalisib) showed 40% or less healthy nuclei, whereas for example the dual PI3K-AKT-mTOR inhibitor omipalisib, known to cause apoptosis in this concentration range [54], as well as the multi-kinase inhibitor TP-0903 [55] and the TRK inhibitor GNF-5837 [56], both known to have an impact on cell viability, decreased the healthy nuclei count more than 60%. Compounds that showed a phenotypic characteristic above the threshold in all three cell lines were ‘flagged’. Further investigations are warranted, if the mode of action is responsible for the ‘flagged’ phenotype or unintended compound features, such as off-target effects or inappropriate concentrations used. In our test chemogenomic set, 49 compounds were ‘flagged’ (Figure 5B). For example, compound KN-62 was marked due to tubulin effects greater 90%. KN-62 is a calcium/calmodulin kinase inhibitor [57], which inhibits the polymerization of tubulin [57], so the phenotypical effect can be explained by its mode of action. For the TIE2 inhibitor BAY-826, there is no link to tubulin function known so far [58]. However, the recently described off-target activity on DDR1/2 may explain the observed phenotype [59]. In total, eight compounds were marked to have tubulin effects while they crossed no other phenotypic threshold (BAY-826, bromosporine, CINK4, PF-299804, SR318, SU11274, YM-201636, ZM447439) in U2OS cells. We detected mitochondrial mass increase for 10 compounds (azelastine, GSK1070916, JNJ-5207787, ML-290, NVP-AEW541, PD 102807, SGC-GAK-1, TC-G 1003, topotecan, XMD17-109) whereas only the pan-HER kinase inhibitor PF-299804 [60] demonstrated membrane permeabilization effects of more than 65%. Importantly, the protocol allows for continuous monitoring enabling the detection of time-dependent observations. For example, WZ-4002, a mutant selective covalent EGFR inhibitor showed initially (12 h) at 1 µM an effect on tubulin and mitochondrial mass before causing membrane permeabilization at 24 h.

## 3. Discussion

Microscopy-based high-content screening, as a strategy for drug discovery, allows monitoring of multiple phenotypes in a fast and economical way [62]. Phenotypic screening has regained attention in drug discovery in recent years. In comparison to target-based drug discovery methods, phenotypic screening does not rely on the knowledge of a specific target *per se* and works as a tool to address complex relations of poorly understood diseases [5]. Extracting information from biological images collected during phenotypic screening and reducing them to a multidimensional profile, a process called image-based profiling, can be used to identify new disease-associated phenotypes, provide a better understanding about target effects and to predict compound activity, toxicity and mechanism of actions [63]. Here, we present HighVia Extend, a live-cell, expandable, unbiased, image-based profiling assay, suitable for real-time measurements [64]. Similar to HighVia, HighVia Extend is modular in nature, inexpensive and flexible, providing the possibility to add additional fluorescent dyes for further readouts or adaptations for the use in different cell lines. Importantly, the assay is applicable for kinetic measurement for over 72 h and can therefore differentiate between primary target effects and secondary phenotypic results caused by the compound treatment. The lack of kinetic information is a frequent problem in phenotypic screens, which monitor endpoints [65]. Using a single readout, Hoechst33342, to assess cell nuclei, we were able to identify healthy cells with high confidence, which enabled the use of additional stains to detect changes in tubulin appearance and mitochondrial content, respectively. Adding the FUCCI system, additional information regarding compounds affecting the cell cycle could be obtained. However, compared to CellPainting, which uses mostly fixed cells and is based on the generation and evaluation of thousands of features [3] our assay provides comprehensive information about cytotoxicity with considerably less features. Thus, the subsequent data processing is less demanding on bioinformatics capabilities while providing additional kinetic aspects. The modular nature of the assay allows for free combination with other dyes or a pre-screening of compounds with only Hoechst33342 and nuclear gating of the cells to reduce the costs of live-cell dyes. We also successfully combined this experiment with other less complex cytotoxic screens as primary screens, such as proliferation experiments using a plate-reader based readout assessing the metabolic state of cells.

The presented assay offers a suitable annotation for (chemogenomic) libraries, providing information on the effect of these compounds on cellular health. It can be used in combination with assays assessing other aspects of cellular health, such as proteome stress involving protein misfolding and aggregation to better annotate a compound library [66]. Our assay thus helps to distinguish between false-positive or false-negative results of subsequent phenotypic assays [67,68]. False negative results can for example be caused by compounds with low solubility or precipitation of a compound as well as low permeability properties. Poorly soluble compounds can also cause false positive results, which may arise by causing unspecific cell death. Another potential source of false negative data might arise due to the missing expression of certain proteins in the tested cell line. The use of several cell lines in parallel as well as assessing the expression profiles using mRNA sequencing databases can, to a certain extent, offset this bias. Other compounds may cause false positive signals in cell assays due to reactivity of structural groups under applied conditions such as redox effects, complex formation, intrinsic fluorescence, degradation and others [68,69]. In the literature, already a large number of small molecules have been annotated as substances to frequently interfere with different assays [70]. Additional unspecific effects on cellular viability have been described for compounds binding to tubulin, e.g., Gul et al. showed that the preclinical used MTH1 inhibitor TH588 showed decreased tumor growth due to involvement in microtubule spindle regulation instead of the first investigated target effect [19,71]. The assessment of the tubulin modulating properties of compounds in a library can thus provide an alert with respect to the downstream effect on cell viability, which is particularly important for cancer cell biology.

For compounds without specific binding information to a protein as well as for target validation, the assay can provide a simple profile for each compound in a time dependent manner. By comparing the effect on cellular health for compounds targeting the same protein, unspecific effects can be easily detected using further analysis and clustering of results. Testing a well-annotated compound collection can thus be used to identify new biology mechanisms for known targets or even find new target correlations.

### 3.1. Materials and Methods HighVia Extend Protocol

For testing the High-Via Extend protocol, nine reference compounds (digitonin, torin, ricolinostat, paclitaxel, staurosporine, JQ1, berzosertib, milciclib, camptothecin) with known mode of actions (Appendix A) were dissolved in DMSO to a concentration of 10 mM. A 7-point serial dilution of every compound were tested in three different cell lines (HEK293T, U2OS, MRC-9). HEK293T (ATCC^®^ CRL-1573™) and U2OS (ATCC^®^HTB-96™) were cultured in DMEM plus L-Glutamine (High glucose) supplemented by 10% FBS (Gibco, Waltham, MA, USA) and Penicillin/Streptomycin (Gibco). MRC-9 fibroblasts (ATCC^®^ CCL-2™) were cultured in EMEM plus L-Glutamine supplemented by 10% FBS (Gibco) and Penicillin/Streptomycin (Gibco). Cells were seeded at a density of 1250 cells per well in a 384 well plates in culture medium (Cell culture microplate, PS, f-bottom, µClear^®^, 781091, Greiner), with a volume of 50 µL per well. All outer wells were filled with 100 µL PBS-buffer (Gibco). Simultaneously with seeding, cells were stained with 60 nM Hoechst33342 (Thermo Scientific, Waltham, MA, USA), 75 nM MitoTracker^TM^ far red (Invitrogen, Waltham, MA, USA), 0.3 µL/well Annexin V Alexa Fluor 488 conjugate (Invitrogen, MA, USA) and 1 µM YoPro3 (Invitrogen, MA, USA).

Cellular shape and fluorescence of the untreated cells was measured 24 h after seeding, using the CQ1 high-content confocal microscope (Yokogawa, Tokyo, Japan). The compounds were added in a 1:1000 dilution (50 nL/well) using an Echo 550 (LabCyte, San Josef, CA, USA) and measured again once and then every 12 h over 72 h. The following setup parameters were used for image acquisition: Ex 405 nm/Em 447/60 nm, 500 ms, 50%; Ex 561 nm/Em 617/73 nm, 100 ms, 40%; Ex 488/Em 525/50 nm, 50 ms, 40%; Ex 640 nm/Em 685/40, 50 ms, 20%; bright field, 300 ms, 100% transmission, one centered field per well, seven z-stacks per well with 55 µm spacing.

Images were analyzed using the CellPathfinder software (Yokogawa), segmented and classified as described previously [22]. Briefly, using an automated algorithm, cell “nuclei” were identified by Hoechst channel intensity levels and optimized by smoothing of mean intensity levels, thresholding and afterwards size-filtering to accurately segment nuclei from cytosol. The ‘cell body’ was defined using the bright field channel. The digital phase contrast was determined between z-stack 3 and 5 with a phase-contrast level of 0.003 to improve cellular shape separation from background. To better identify cytoplasmic areas, the threshold results of the cell body were defined as interdependent of nuclei. After segmentation of the cells, classification was performed using the machine learning feature of the CellPathfinder Software. Training of the machine learning algorithm was performed by an experienced cell biologist. The cells were classified in healthy, early apoptotic, late apoptotic, necrotic and lysed cells by 19 features of the cell body and 13 features of the nuclei (Appendix A including dye intensity levels and cellular morphology characteristics such as cell diameter or compactness. Different control compounds were used to train the machine learning algorithm. Staurosporine 10 µM was used to identify apoptotic cells, digitonin 10 µM was used to classify lysed cells. The analysis was validated using duplicate wells of the named compounds. For nuclei classification, the cells were subdivided in healthy, pyknosed and fragmented nuclei by ten features (Appendix A) of the Hoechst channel. To detect objects that show high intensity of the Hoechst channel, classification in High Intensity Objects and Normal Intensity Objects was implemented using three features for the cell body and two features for the nuclei (Appendix A). The health cell count and the healthy nuclei count were normalized against the healthy cell count and healthy nuclei count of cells treated with 0.1% DMSO. Significance was calculated using a two-way ANOVA analysis in GraphPad PRISM 8.

### 3.2. Multiplex Protocol

HEK293T (ATCC^®^ CRL-1573™) and U2OS (ATCC^®^HTB-96™) were cultured in DMEM plus L-Glutamine (High glucose) supplemented by 10% FBS (Gibco) and Penicillin/Streptomycin (Gibco). MRC-9 fibroblasts (ATCC^®^ CCL-2™) were cultured in EMEM plus L-Glutamine supplemented by 10% FBS (Gibco) and Penicillin/Streptomycin (Gibco). One day prior to compound exposure, cells were stained simultaneously to seeding with 60 nM Hoechst33342 (Thermo Scientific, MA, USA), 75 nM Mitotracker red (Invitrogen, MA, USA), 0.3 µL/well Annexin V Alexa Fluor 680 conjugate (Invitrogen, MA, USA) and 25 nL/well BioTracker™ 488 Green Microtubule Cytoskeleton Dye (EMD Millipore, MA, USA). Cells were seeded at a density of 2000 cells per well in a 384 well plates in culture medium (Cell culture microplate, PS, f-bottom, µClear^®^, 781091, Greiner, Frickenhausen, Germany), with a volume of 50 µL per well. All outer wells were filled with 100 µL PBS-buffer (Gibco).

Using the CQ1 high-content confocal microscope (Yokogawa), cellular shape and fluorescence was measured before and 12 h as well as 24 h after compound treatment. All compounds were diluted in DMSO to a concentration of 10 mM. Compounds were added directly to the cells in a 1:1000 dilution (50 nL/well) using an Echo 550 (LabCyte, San Josef, CA, USA).

For image acquisition, the following parameters were used: Ex 405 nm/Em 447/60 nm, 500 ms, 50%; Ex 561 nm/Em 617/73 nm, 100 ms, 40%; Ex 488/Em 525/50 nm, 50 ms, 40%; Ex 640 nm/Em 685/40, 50 ms, 20%; bright field, 300 ms, 100% transmission, one centered field per well, seven z-stacks per well with a total of 55 µm spacing. The rather large spacing distance was used to create a robust readout, compensating potential plate variations and enabling automated screening without the use of autofocus. The overlap of the fluorescence emission spectra of the dyes was neglectable for all but the MitoTracker Red and Annexin V Alexa Fluor 680 (Appendix A). However, this overlap does not influence the analysis, since the excitation maxima of these two dyes are well separated and the gating algorithm analyses only the Mitotracker Red intensity in Annexin 5 negative cells.

All images were analyzed using the CellPathfinder software (Yokogawa). Segmentation of cells was performed as described earlier. First, the cells are classified in Hoechst High Intensity Objects or Normal Intensity Objects (Appendix A). All normal gated cells are further classified in healthy, fragmented or pyknosed nuclei (Appendix A). The pyknosed cells are gated in mitotic or apoptotic cells using seven features for the cell body and five features for the cell nuclei according to their Annexin V staining intensity (Appendix A). All cells that were classified as including a healthy nucleus are further gated into three phenotypic classes. They are gated in tubulin effect or no tubulin effect (Appendix A), mitochondrial mass increased or not increased (Appendix A) and membrane permeability/membrane normal (Appendix A). Growth rate was calculated against non-treated cells and cells treated with DMSO 0.1% [61].

### 3.3. FUCCI Assay Protocol

For generation of a stable cell line, including the florescent ubiquitination-base cell cycle indicator FUCCI, the plasmid-based transposon system Sleeping Beauty was used. Vector (pSBbi_Fucci) and the Transposase SB100X have been described previously [72]. HCT116 cells (ATCC^®^ CCL-247™) were cotransfected in a small cell culture flask (5 mL) with a mixture of 9.5 µg pSBbi_Fucci vector and 0.5 µg if the SB100X transposase vector using FuGENE HD (Promega) as described previously [73]. Two days after transfection, cells were selected over 10 days using puromycin (1 µg/mL) and cultivated afterwards for two more weeks in McCoys 5A plus L-Glutamine (Gibco) supplemented by 10% FBS (Gibco) and Penicillin/Streptomycin (Gibco).

HCT116-FUCCI cells were seeded at a density of 1250 cells per well in a 384 well plate (Cell culture microplate, PS, f-bottom, µClear^®^, 781091, Greiner) in culture medium to 50 µL per well and stained additionally with 60 nM Hoechst33342 (Thermo Scientific). Outer wells were filled with 100 µL PBS-buffer (Gibco). Fluorescence and cellular shape were measured before and after compound treatment for 72 h every 12 h using the CQ1 high-content confocal microscope (Yokogawa). Compounds were added directly to the cells, using an Echo 550 (LabCyte, San Josef, CA, USA) in a 1:1000 dilution (50 nL/well) to a final concentration of 10 µM.

Following parameters were used for image acquisition: Ex 405 nm/Em 447/60 nm, 500 ms, 50%; Ex 561 nm/Em 617/73 nm, 100 ms, 40%; Ex 488/Em 525/50 nm, 50 ms, 40%; Ex 640 nm/Em 685/40, 50 ms, 20%; bright field, 300 ms, 100% transmission, one centered field per well, seven z-stacks per well with a total of 55 µm spacing. Image analysis was performed using the CellPathfinder software (Yokogawa). Segmentation of cells was performed as described earlier. First, the cells are classified in Hoechst High Intensity Objects or Normal Intensity Objects (Appendix A). All normal gated cells are further classified in healthy, fragmented or pyknosed nuclei (Appendix A). The cells that showed healthy nuclei were then further gated in red/green or yellow using 11 features of the cell body and four features of the cell nuclei (Appendix A). Total cell count was normalized against total cell count of cells treated with 0.1% DMSO.

### 3.4. Dye Titration CQ1 and Alamarblue Assay

U2OS cells (ATCC^®^HTB-96™) were cultured in DMEM plus L-Glutamine (High glucose) supplemented by 10% FBS (Gibco) and Penicillin/Streptomycin (Gibco) and seeded on a 384 well plate in culture medium (Cell culture microplate, PS, f-bottom, µClear^®^, 781091, Greiner, Frickenhausen, Germany) with half of the plate with and the other half of the plate, without dyes at a density of 1500 cells per well in a, in a volume of 50 µL per well. All outer wells were filled with 100 µL PBS-buffer (Gibco). Then, 24 h after seeding, cells without dyes were treated with the same concentration of dyes and directly measured by the CQ1 confocal microscope (Yokogawa) over 72 h every 12 h. The dyes Hoechst33342 (Thermo Scientific, MA, USA), Mitotracker red (Invitrogen, MA, USA), Mitotracker far red (Invitrogen, MA, USA), Annexin V Alexa Fluor 680 conjugate (Invitrogen, MA, USA), Annexin V Alexa Fluor 488 conjugate (Invitrogen, MA, USA), BioTracker™ 488 Green Microtubule Cytoskeleton Dye (EMD Millipore, Massachusetts, United States), YoPro3 (Invitrogen, MA, USA)) were added using an Echo 550 (LabCyte, San Josef, CA, USA) in a 7-fold dilution and different dye combinations (Appendix A). Image acquisition was completed with the following parameters: Ex 405 nm/Em 447/60 nm, 500 ms, 50%; Ex 561 nm/Em 617/73 nm, 100 ms, 40%; Ex 488/Em 525/50 nm, 50 ms, 40%; Ex 640 nm/Em 685/40, 50 ms, 20%; bright field, 300 ms, 100% transmission, one centered field per well, seven z-stacks per well with a total of 55 µm spacing. Image analysis was performed using the CellPathfinder software (Yokogawa) as described earlier. To detect the cells without Hoechst33342 stain, the cell body was defined just by bright field intensity levels. Cells were classified using machine learning algorithms by an experienced cell biologist as healthy or not healthy.

After 72 h, the plate was treated with 1:10 alamarBlue™ (ThermoFisher, MA, USA) solution for 12 h. Fluorescence was measured on a PHERAstar plate reader (BMG Labtech, Ortenberg, Germany) with an emission of 590 nm and excitation of 545 nm.

## Figures and Tables

**Figure 1 molecules-27-01439-f001:**
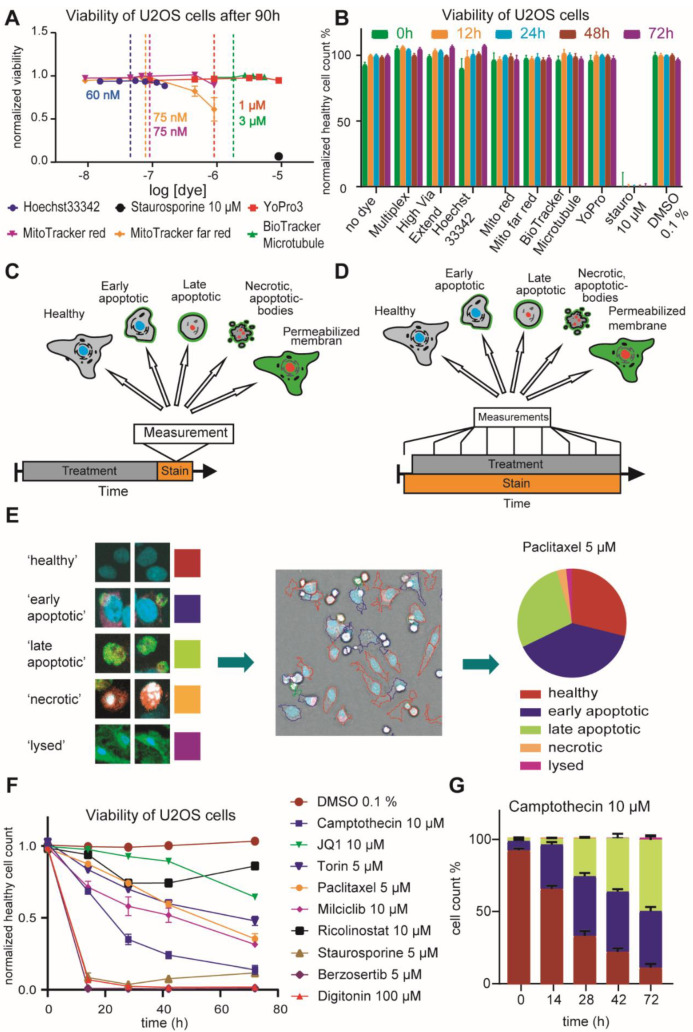
Validation of cell staining dyes and optimization of High-Via Protocol. (**A**) Healthy cell count after 90 h of cell staining dye exposure at six different concentrations and 10 µM of staurosporine in U2OS cells. Vertical lines show optimal concentration used in subsequent assays (Hoechst33342 60 nM (ThermoFisher, Waltham, MA, USA), MitoTracker ^TM^ red 75 nM (Invitrogen, Waltham, MA, USA), MitoTracker ^TM^ far red 75 nM (Invitrogen, Waltham, MA, USA), YoPro3 1 µM (Invitrogen, Waltham, MA, USA), BioTracker Microtubule 3 µM (EMD Millipore, Burlington, MA, USA). Error bars show SEM of four technical replicates. (**B**) Healthy cell count 0 h, 12 h, 24 h, 48 h and 72 h after cell staining dye exposure alone or in combination (Multiplex: Hoechst33342 60 nM, BioTracker^TM^ Microtubule 3 µM, MitoTracker ^TM^ red 75 nM, Annexin V Alexa Fluor 680 0.3 µL/well; HighVia Extend: Hoechst33342 60 nM, MitoTracker ^TM^ far red 75 nM, YoPro3 1 µM, Annexin V Alexa Fluor 488 0.3 µL/well) and 10 µM of staurosporine (stauro) normalized to healthy cells exposed to DMSO 0.1% in U2OS cells. Error bars show SEM of four technical replicates. (**C**) Generic workflow of cell gating in HighVia protocol. (**D**) Generic workflow of cell gating in HighVia Extend protocol. (**E**) Cellular classification in healthy, early apoptotic, late apoptotic, necrotic and lysed by trained cell biologist. Cellular classifications shown after segmentation using a machine learning algorithm. Fractions of healthy, early apoptotic, late apoptotic, necrotic and lysed cells after 24 h of 5 µM paclitaxel exposure in U2OS cells. (**F**) Healthy cell count before and 14 h, 28 h, 42 h and 72 h after compound exposure (camptothecin 10 µM, JQ1 10 µM, torin 5 µM, paclitaxel 5 µM, milciclib 10 µM, ricolinostat 10 µM, staurosporine 5 µM, berzosertib 5 µM, digitonin 100 µM) normalized to healthy cells exposed to DMSO 0.1% in U2OS cells. Error bars show SEM of technical triplicates. (**G**) Fractions of healthy, early apoptotic, late apoptotic, necrotic and lysed cells before and 14 h, 28 h, 42 h and 72 h after 10 µM camptothecin exposure in U2OS cells. Error bars show SEM of technical triplicates.

**Figure 2 molecules-27-01439-f002:**
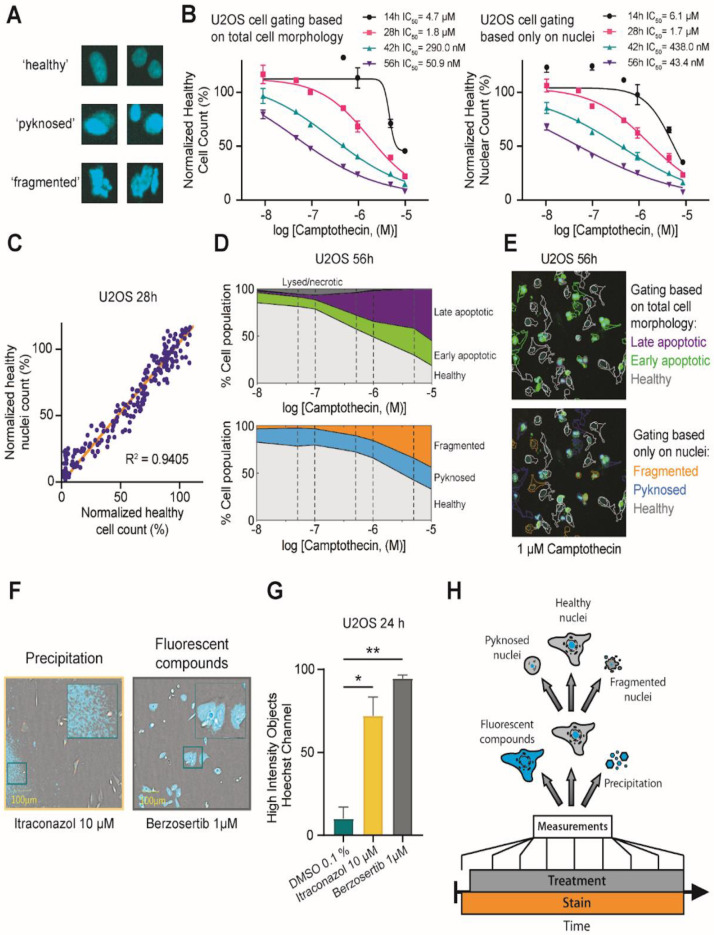
Analysis of Cell Nuclei by Hoechst Channel Intensity level. (**A**) Nuclei classification in healthy, pyknosed and fragmented by trained cell biologist. (**B**) Normalized healthy cell count and normalized healthy nuclear count of different concentrations (0.01 µM, 0.05 µM, 0.1 µM, 0.5 µM, 1 µM, 5 µM, 10 µM) of camptothecin exposure with calculated IC50 values after 14 h, 28 h, 42 h and 56 h. (**C**) Correlation between the healthy cell count and the healthy nuclei count after 28 h of compound exposure normalized to healthy cells exposed to DMSO 0.1% in U2OS cells. (**D**) Fractions of HighVia gating and fractions of healthy, fragmented and pyknosed nuclei after exposure to different concentrations (0.01 µM, 0.05 µM, 0.1 µM, 0.5 µM, 1 µM, 5 µM, 10 µM of camptothecin in U2OS cells after 56 h of compound exposure. Error bars show SEM of three technical replicates. (**E**) Fluorescence image of different gatings of U2OS cells exposed to 1 µM camptothecin after 56 h based on total cell morphology and based on nuclei. (**F**) Bright field confocal image of stained (blue: DNA/nuclei, green: Annexin V apoptosis marker, red: YoPro3, magenta: Mitotracker Deep Red, mitochondria content) U2OS cells after 24 h of compound exposure (itraconazole 10 µM, berzosertib 1 µM). Precipitation of 10 µM itraconazole and fluorescence of 1 µM berzosertib exposure shown as Hoechst High Intensity Objects are highlighted. (**G**) Hoechst High Intensity Objects after 24 h of compound exposure (itraconazole 10 µM, berzosertib 1 µM) and DMSO 0.1% as negative control. Error bars show SEM of three technical replicates. (**H**) Generic workflow of cell gating based on cell nuclei.

**Figure 3 molecules-27-01439-f003:**
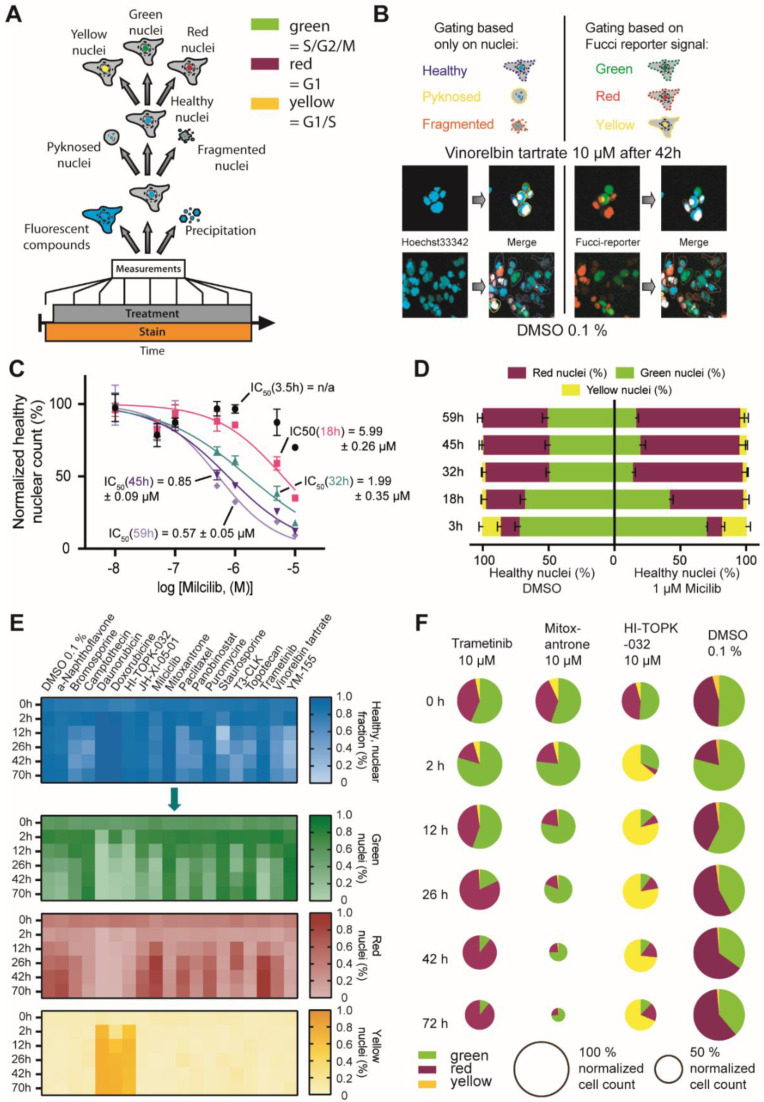
FUCCI Cell Cycle Assay. (**A**) Gating scheme of the machine learning-based analysis of the HCT116-FUCCI reporter cell line (**B**) Example images of HCT116 cells treated either with 10 µM vinorelbine tartrate or 0.1% DMSO for 42 h. Highlighted by colored lines are the gating results of both the nuclear gating step into “healthy”, “pyknosed” and “fragmented” nuclei and the FUCCI gating step of ‘healthy’ nuclei only. (**C**) Cytotoxicity of milciclib for HCT116-FUCCI at different time points assessed by the normalized healthy cell count. The depicted IC50 values are the average and SEM of two independent experiments. (**D**) Population distribution of ‘healthy’ gated (based on nuclear features) HCT116-FUCCI cells treated with 1 µM milciclib or 0.1% DMSO for different periods. (**E**) Heat map analysis of HCT116-FUCCI cells treated with 10 µM of different inhibitor for different time points. (**F**) Detailed representation of the population fractions of HCT116-FUCCI cells exposed to 10 µM of trametinib, mitoxantrone and HI-TOPK-032. The size of the pie charts reflected the normalized cell count. For comparison, the fractions of DMSO are shown as well.

**Figure 4 molecules-27-01439-f004:**
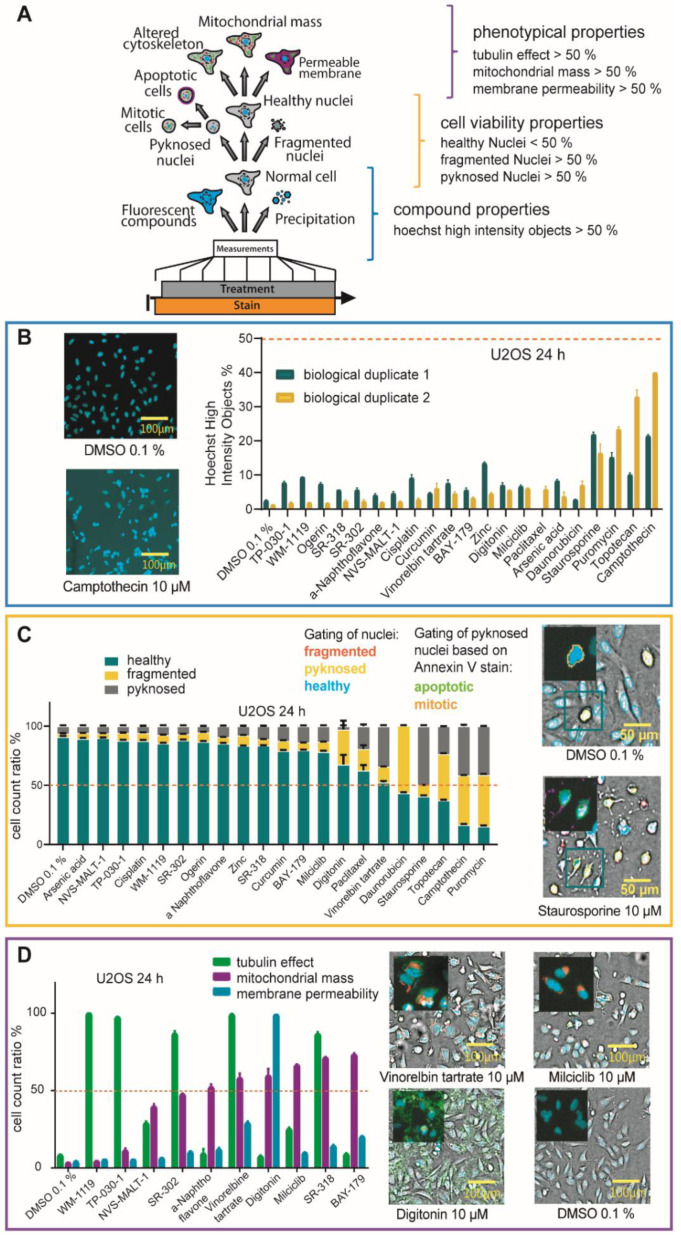
High Content Multiplex screen of different compounds in U2OS cells. (**A**) General workflow of Multiplex High Via protocol analysis with property thresholds. (**B**) Ratio of Hoechst High Intensity Objects after 24 h of compound exposure (Appendix A) in U2OS in comparison to DMSO 0.1% of biological duplicates. Error bars show SEM of technical triplicates. Property threshold at 50% marked red. For example, fluorescence confocal images of stained (blue:DNA/nuclei, Hoechst33342) U2OS cells after 24 h of exposure to camptothecin 10 µM in comparison to DMSO 0.1%. (**C**) Cell count ratio of different Nuclei gating after 24 h of 10 µM of compound exposure (Appendix A) in comparison to DMSO 0.1% in U2OS cells. Error bars show SEM of technical triplicates. Property threshold at 50% marked red. Brightfield confocal image of stained (blue: DNA/nuclei, green: Microtubule different, red: mitochondria content, magenta: Annexin V apoptosis marker) U2OS cells after 24 h of compound exposure of staurosporine 10 µM) in comparison to DMSO 0.1%. Gating of cells for Nuclei gating and Annexin V based gating shown. (**D**) Cell count ratio of tubulin effect (green), mitochondrial mass increase (magenta) and membrane permeability (blue) of U2OS cells after 24 h of 10 µM of compound exposure (WM-1119, TP-030-1, NVS-MALT-1, SR-302, α-naphthoflavone, vinorelbine tartrate, digitonin, milciclib, SR-318, BAY-179) in comparison to DMSO 0.1%. Error bars show SEM of technical triplicates. Property threshold at 50% marked. Brightfield confocal image of stained (blue: DNA/nuclei, green: microtubule different, red: mitochondria content, magenta: Annexin V apoptosis marker) U2OS cells after 24 h of compound exposure (vinorelbine tartrate 10 µM, milciclib 10 µM, digitonin 10 µM) in comparison to DMSO 0.1%. Fluorescent image of different staining is highlighted.

**Figure 5 molecules-27-01439-f005:**
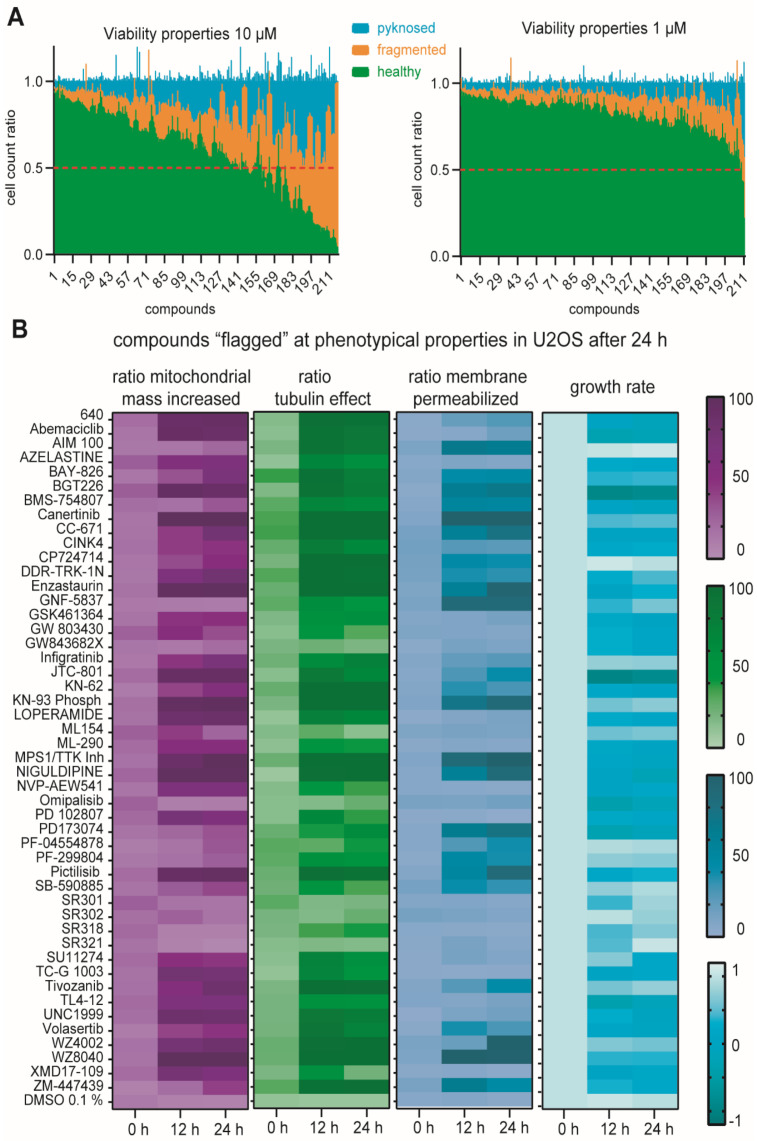
Multiplex Certificates of Chemogenomic Compounds. (**A**) Cell count ratio of different nuclei gating after 24 h of 10 µM and 1 µM of compound exposure (Appendix A) in U2OS cells. Error bars show SEM of biological duplicates. Healthy nuclei count at 50% (viability threshold) marked as a red line. (**B**) Heat map of phenotypical property ratios (tubulin effect, mitochondrial mass increased and membrane permeabilized) and the growth rate, which was calculated against the non-treated cell number, as described earlier by Hafner et al. [61] of U2OS cells exposed to 49 compounds, that were marked as ‘flagged’ after Multiplex analysis (phenotypical property threshold > 50%). Heat map shows meaning of two biological duplicates. All data are available in Appendix A.

## Data Availability

All data used in this study is available in the BioImage Archive under https://www.ebi.ac.uk/biostudies/studies/S-BIAD145 (accessed on 2 January 2022).

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
