# Peer review of "Image-Based Annotation of Chemogenomic Libraries for Phenotypic Screening"

_molecules, 2022, doi:10.3390/molecules27041439_

Round 1

Reviewer 1 Report

The Authors describe and expanded panel of probes from their earlier work (SLAS Discovery 2020) on the HiVia protocol towards investigating and characterizing the cytotoxic mechanism of small molecules.  A nice addition to this updated protocol has been the inclusion of additional cellular health markers, temporal data, and the multiplexing ability.  This could help in catching and characterizing many of the oncology directed probes that may induce cell death but not via their intended target/mechanism (Lin, A. et al. Sci. Transl. Med. 11, eaaw8412 (2019)).  The HiVia Extended method is compared to Cell Painting and does offer simplicity in not necessitating the complex staining antibody staining protocol, static fixed cellular imaging, and complex analysis.  The HiVia however might be better compared to FluoPak (Kang, Z. et al. Sci Reports 10, 2097 (2020)) with similarities in LMW dyes and temporal readout.  

A few minor points for consideration would be the potential target expression in the cell lines profiled for the on-and off-targets of the profiled  compounds may lead to variable or unanticipated results.  Additionally it would be interesting as to how HiVia might be expanded to other cell lines that may be more sensitive towards compound treatment and/or more challenging to image than U2OS, HEK293T, etc cells.  

I think updated extended version of HiVia and the multiplexing capabilities are solid additions and worthy of publication in Molecules with minor revisions.

A few minor spelling errors were found i.e. Figure 4 digitonin and puromycin.

Author Response

Answer: We thank the reviewer for the positive remarks on the mansucript. We agree with the reviewer that cell-line specific expression patterns can influence chemogenomic experiments. We therefore tested the protocol against three different human cell lines of various origin (U2OS, HEK293T and MRC-9) to decrease the risk of only detecting cell-line dependent effects. However, we agree that one should confirm the expression pattern for individual proteins and thus added the following sentence to the discussion: “Another potential source of false negative data might arise due the missing expression of certain proteins in the tested cell line. The use of several cell lines in parallel as well as assessing the expression profiles using mRNA sequencing databases can, to a certain extent, offset this bias.”

We also agree with the reviewer that it would be interesting for the reader to see that the described assay can also be expanded to more challenging cell lines. The fibroblast cell line MRC-9 shows a distinct cellular morphology compared to U2OS or HEK293T cells with a less globular shape and a different nucleus/cytoplasm ratio. The analyses for fibroblasts were as robust as for the other two cell lines shown in Supplementary Figure S5, S6 and S8. Nevertheless, we agree that not all cells, such as non-adherent cells, are suitable for the decsribed method.

Additionally, we analyzed the data of the original HighVia paper performed on HeLa cells in high confluency to test further the limits of the cellular detection and gating algorithms. Similar to the data of U2OS, HEK293T and MRC-9, we observed also for HeLa cells a strong correlation between non/healthy cells and altered nuclear phenotypes (see attached Figure)

We have amended the spelling mistakes.

Reviewer 2 Report

The manuscript by Tjaden et al., entitled “Image based annotation of Chemogenomic Libraries for Pheno-1 typic Screening” describes a modular kinetic version of their previously published HighVia protocol for assess cell health effects by compounds.  The paper presents optimized dye concentrations to provide a robust kinetic protocol, as well as insights into using a single dye  - Hoechst33342 nuclear stain alone to flag artifacts (fluorescent or insoluble compounds) and phenotypes. Overall it is well-written and will be of interest to readers. A few comments:

  1. It is noted in the Discussion that the HighVia extend protocol “requires less extensive data storage and bioinformatics capabilities” compared to Cell Painting. It would seem that the kinetic aspect, number of cell stains, and use of machine learning in the presented method would not save much – can the authors be more specific about this point?
  2. The authors have provided a good deal of information on using imaging to determine cell health effects for compounds including the previous single time point HighVia method. Can the authors offer some advice in the Discussion on an optimal workflow for these assays – e.g. should one first panel using Hoechst alone and bin compounds/flag artifacts then move to higher multiplex assays?

Minor

  1. Germen spelling of “before” is used in the Materials and Methods (“bevor”)
  2. Was the z-stack interval really 55um? This seems a bit large for 7 z stacks. Was a maximum intensity projection used for the image analysis or how were the z stacks treated?

Author Response

1. Answer: We thank the referee for the positive feedback on the manuscript and we agree that the sentence was not clearly written and we therefore rephrased this part to highlight better the computational differences of Cell Painting with the here described method.

“However, compared to CellPainting, which uses mostly fixed cells and is based on the generation and evaluation of thousands of features [3] our assay provides comprehensive information about cytotoxicity with considerable less features. Thus, the subsequent data processing is less demanding on bioinformatics capabilities while providing additional kinetic aspects.”

2. Answer: For a less expensive and easier workflow a first approach based only on Hoechst can be used and we have therefore included the following sentence in the discussion:

“The modular nature of the assay allows for free combination with other dyes or a pre-screening of compounds with only Hoechst33342 and nuclear gating of the cells to reduce the costs of live-cell dyes. We also successfully combined this experiment with other less complex cytotoxic screens as primary screens, such as proliferation experiments using a plate-reader based readout assessing the metabolic state of cells.”

German spelling of “before” is used in the Materials and Methods (“bevor”)

Answer: We thank the reviewer for pointing out the mistakes and amended those accordingly.

Was the z-stack interval really 55um? This seems a bit large for 7 z stacks. Was a maximum intensity projection used for the image analysis or how were the z stacks treated?

Answer: We have included a short explanation in the methods and materials part why this distance of the z stacks was used as the protocol was optimised towards robustness:

“The rather large spacing distance was used to create a robust readout, compensating potential plate variations and enabling automated screening without the use of autofocus.”

Reviewer 3 Report

Tjaden et al. present a systematic work describing a protocol to use image based phenotypic screening. This protocol relies on different cell stains for fluorescence high content imaging. They first validated different staining dyes' impact on cell viability. Followed by nuclear analyses, they focused on the cell cycle analysis. Finally, they utilized the multiplex protocol to provide expanded profiles of small molecule drug hits based on cellular parameters.

This work was well written and the figures were well designed and very attractive to authors. I support its publication after the author may consider the following suggestion.

  1. Have the authors considered the potential fluorescence signal crosstalk due to the large bandpass of the high content imagers and the overlap of fluorescence emission spectra of different dyes?
  2. Health of the cell may not only correlate to cell viability. Stresses such as proteome stresses, may also contribute to the health of the cell but showed no effect on cell viability assay. The authors may consider to elaborate the definition of this term. The author may refer to this article Angew Chem Int Ed Engl, 2017, 56, 8672-8676 regarding this issue.

Author Response

1. Answer: We thank the reviewer for the positive feedback and remarks on the manuscript and included a short explanation on the posibility of crosstalk and overlay of the different fluorescent emission spectra of the different dyes in the methods and materials part and have added a new Supplementary Figure (S9):

“The overlap of the fluorescence emission spectra of the dyes was neglectable for all but the MitoTracker Red and Annexin V Alexa Fluor 680 (Supplementary Figure S9). However, this overlap does not influence the analysis, since the excitation maxima of these two dyes are well separated and the gating algorithm analyses only the Mitotracker Red intensity in Annexin 5 negative cells”. Additionally, we added Supplementary Figure S9 to show the emission spectra using the Thermo Fisher Fluorescnece Spectra Viewer (https://www.thermofisher.com/uk/en/home/life-science/cell-analysis/labeling-chemistry/fluorescence-spectraviewer.html).

2. Answer: We have added a sentence pointing out that cell health does not only refer to cell viability, including the suggested article. We thank the reviewer for pointing out the work on proteom stress.